# Dynamic Regulation of Lipid Droplet Biogenesis in Plant Cells and Proteins Involved in the Process

**DOI:** 10.3390/ijms24087476

**Published:** 2023-04-19

**Authors:** Yiwu Zhao, Qingdi Dong, Yuhu Geng, Changle Ma, Qun Shao

**Affiliations:** Shandong Provincial Key Laboratory of Plant Stress, College of Life Sciences, Shandong Normal University, Jinan 250358, China

**Keywords:** lipid droplets, biogenesis, triacylglycerol synthesis, nucleation, LD budding, LD coat formation

## Abstract

Lipid droplets (LDs) are ubiquitous, dynamic organelles found in almost all organisms, including animals, protists, plants and prokaryotes. The cell biology of LDs, especially biogenesis, has attracted increasing attention in recent decades because of their important role in cellular lipid metabolism and other newly identified processes. Emerging evidence suggests that LD biogenesis is a highly coordinated and stepwise process in animals and yeasts, occurring at specific sites of the endoplasmic reticulum (ER) that are defined by both evolutionarily conserved and organism- and cell type-specific LD lipids and proteins. In plants, understanding of the mechanistic details of LD formation is elusive as many questions remain. In some ways LD biogenesis differs between plants and animals. Several homologous proteins involved in the regulation of animal LD formation in plants have been identified. We try to describe how these proteins are synthesized, transported to the ER and specifically targeted to LD, and how these proteins participate in the regulation of LD biogenesis. Here, we review current work on the molecular processes that control LD formation in plant cells and highlight the proteins that govern this process, hoping to provide useful clues for future research.

## 1. Introduction

Lipid droplets (LDs) are ubiquitous organelles which contain a hydrophobic lipid core surrounded by a phospholipid monolayer. The cell biology of LDs has been the focus of intensive research over the past decade because of their functions in a wide range of developmental processes [1]. In eukaryotes, LDs are derived from the endoplasmic reticulum (ER). However, the mechanisms underlying LD biogenesis had remained incompletely understood until recently. LD biogenesis is a highly coordinated and stepwise process involving different sets of organism-specific LD packaging proteins [2]. Several models of LD biogenesis have been proposed [3,4,5,6], among which the conventional model of oil formation at the ER and budding into the cytosol is the most widely accepted. According to this model, LD biogenesis is divided into three main steps: nucleation, growth and budding. Nucleation refers to the initial accumulation of neutral lipids (mainly triacylglycerols (TAGs)) which form a lipid “lens” between the two ER monolayers. Growth consists of size expansion due to synthesized TAG absorption or fusion of nascent small LDs. Budding is the process by which the ER membrane leaflet bulges into the cytoplasm, eventually budding off to form the nascent LD [7,8]. The precise mechanisms of nascent LD emergence and growth, and its preference for directional budding, are still unclear [9,10].

Understanding of the LD biogenesis process mainly comes from research in animal and yeast systems. In plants, however, understanding of LD formation is more fragmented and the mechanistic details of LD formation remain poorly defined. As in other organisms, in plants, LDs are also thought to emerge from the ER. Most experimental studies on LD formation in plants have been performed in developing oilseeds. Although neutral lipid accumulation is conserved functionally between organisms, other aspects of LD biogenesis in plants, especially the LD coat, may be performed by distinctly different sets of proteins (Figure 1).

Homologs of some regulatory proteins related to animal LD formation have been identified in plants, however, there is no experimental evidence to show whether they function the same way in plant LD formation [11,12,13]. It has been shown that plant and animal LD surface proteins are completely different (Figure 1). Moreover, the homologs of key proteins regulating animal LD biogenesis, such as fat storage-inducing transmembrane (FIT) proteins [14], perilipin and cell death-inducing DNA fragmentation factor 45-like effector (CIDE) proteins [15], have not yet been found in plants, indicating that in plants other proteins may perform similar functions. Together these data suggest there may be significant differences in the details of how LDs forms in plant cells compared to animals and yeast. Our focus here is to review the current knowledge of the molecular processes that control LD formation in plants, highlighting the proteins that govern this process. We propose a model for LD biogenesis in plants.

## 2. Triacylglycerol (TAG) Synthesis

LD biogenesis in eukaryotic cells begins with the synthesis of neutral lipids—TAGs and/or steryl esters (StEs). In mammals, yeasts and plants, different types of LDs with different surface or internal components coexist, but the initial biophysical processes to form TAG- and StEs-containing LDs are likely similar [16]. So far, most work on LD biogenesis has focused on TAG-containing LDs.

In various plant tissues, fatty acids are biosynthesized in plastids and stored in LDs mainly in the form of TAGs. Most studies on lipid accumulation into LDs have focused on seed tissues. TAGs accumulate in the embryo and endosperm [17] and can comprise as much as 60% of seed weight. At the beginning of oil accumulation, the newly formed acyl carrier protein (ACP) esters of fatty acids are first hydrolyzed by two different types of acyl ACP thioesterases (FATA and FATB, being responsible for hydrolysis of unsaturated and saturated acyl-ACPs, respectively) at the inner plastid envelope membrane [18]. Unesterified fatty acids are then transported to the ER by members of the fatty acid export protein (FAX) family. In *Arabidopsis*, this family has seven members, two of which (FAX2 and FAX4) are highly expressed in early seed development. Unesterified fatty acids shuttle across the plastid outer membrane, possibly through acylation by long-chain acyl coenzyme A (CoA) synthetase (LACS). LACS enzymes catalyze fatty acyl chains to fatty acyl CoAs to form CoA esters, which is the critical activating reaction of TAG biosynthesis. Nine LACS enzymes have been identified in *Arabidopsis*. LACS1 and LACS9 are localized in the ER and plastid, resepectively, and are considered to be the major LACSs involved in TAG biosynthesis [10]. The generated fatty acid CoA esters then enter the TAG synthesis pathway as substrates.

In plants, TAG-synthesizing enzymes are localized to the extended regions or subdomains of the ER, suggesting that TAGs are synthesized primarily in the ER. TAGs consist of a three-carbon glycerol backbone (numbered sn-1, 2 and 3) esterified to fatty acids (FAs) at the three sites. Three main TAG biosynthesis pathways exist: the sn-glycerol-3-phosphate (G3P) or Kennedy pathway, the dihydroxyacetone phosphate pathway and the monoacylglycerol pathway [19,20,21]. In plants, the most important process to create TAGs is the Kennedy pathway, an evolutionarily conserved process found in most living organisms. This pathway consists of the sequential acylation and subsequent dephosphorylation of G3P. First, catalyzed by a G3P acyltransferase (GPAT), G3P is esterified by a fatty acid CoA ester at position sn-1 to form lysophosphatidic acid (LPA). There are 10 members of the GPAT family in *A. thaliana*, among which GPAT9 plays an important role in TAG biosynthesis [22]. LPA is in turn acylated by an LPA acyltransferase (LPAAT) at position sn-2 to generate phosphatidic acid (PA) [23]. Five *LPAAT* genes have been identified in *Arabidopsis* and *LPAAT2* is likely a major contributor to TAG accumulation [24]. Then, PA is converted to diacylglycerol (1,2-diacyl-sn-glycerol, DAG) by removal of the sn-3 phosphate group by PA phosphohydrolases (PAPs). Plants contain multiple PAP isoforms, and two genes (At*PAH1* and At*PAH2*) encoding PAP enzymes exist in *Arabidopsis*. AtPAH1/2 mainly play a role in repressing phospholipid synthesis at the ER [25,26]. AtPAH1 displays higher enzyme activity than AtPAH2. Individual expression of AtPAH1 or AtPAH2 can rescue the phenotype of the yeast *pah1∆* mutant, including severe DAG and TAG deficiencies [27]. These data suggest that AtPAH1/2 also play a role in de novo DAG synthesis in plants.

The esterification of DAG to TAG is the final step in TAG biosynthesis. This process can occur by mechanisms either dependent or independent of acyl CoA esters. In the Kennedy pathway, the acylation of DAG at the sn-3 position is catalyzed by diacylglycerol acyltransferase (DGAT) in a mechanisms dependent on acyl CoA esters [28]. This reaction is one of the rate-limiting steps in lipid accumulation [29]. There are three distinct DGAT enzymes in plants, which have no sequence homology with each other. DGAT1 and DGAT2 are two ER transmembrane-bound isoenzymes with a characterized C-terminal pentapeptide ER retrieval motif. They are unrelated and differ in their membrane topology and substrate discrimination. The two DGATs are also present in LDs and plastids [30,31]. DGAT3 is a soluble, cytosolic form reported in peanut [32] and *Arabidopsis* [33]. In *Arabidopsis thaliana*, DGAT1 is a key enzyme involved in TAG formation in developing seeds [34]. DGAT2 is especially important in plant species with unusual fatty acid compositions [35], while DGAT3 regulates the acyl-CoA pool size and composition according to the needs of membrane lipid biosynthesis during seed development. Recent studies have shown that DGAT3 is a metalloprotein with the ability to synthesize TAG in vitro [36].

Another major enzyme involved in seed TAG accumulation is phospholipid—diacylglycerol acyltransferase (PDAT1). *Arabidopsis* PDAT catalyzes the acyl-CoA-independent pathway and is thought to play a critical role in TAG synthesis in leaves [37]. This enzyme transfers an acyl moiety from the sn-2 position of phosphatidylcholine (PC) to the sn-3 position of DAG to form TAG molecules. PDAT1 and DGAT1 play partially redundant functions in seeds and plants vary in their preference for DGAT or PDAT to produce TAG [38].

At the transcriptional level, lipid synthesis and partition are regulated by several key transcription factors, including WRINKLED1 (WRI1) and LEAFY COTYLEDON2 (LEC2) [39]. TAG content in transgenic *Lemna japonica* fronds overexpressing *AtWRI1* was significantly increased by about 7-fold compared to WT, indicating that WRI1 is a positive regulator of lipid synthesis [40]. The ectopic expression of a seed-specific LEC2 promotes accumulation of TAGs in leaf tissues. This accumulation is accentuated when DGAT or oleosins are also present [41,42].

It is worth noting that SEs are also formed in the ER in yeast [43] and mammals [44,45]. There are considerably fewer studies concerning LDs containing SEs compared with those on LDs containing TAGs in plant and thus are not reviewed in this article. Furthermore, a variety of other hydrophobic molecules have been found to accumulate in LDs, including *O*-acyl ceramide [46], squalene, vitamin E and long-chain polyprenols. However, whether they are capable of triggering LD formation is unknown.

## 3. Nucleation of LD

Nucleation is the initial accumulation of TAGs in the ER membrane where stable nascent LDs will form. After synthesis, individual TAG molecules may freely diffuse in the ER bilayer, randomly meet and assemble. Upon reaching a critical concentration, neutral lipids nucleate and form lens-like structures in the ER membrane [7]. Whether this step is spontaneous, enzymatically controlled or a combination of both is not entirely clear. However, the demixing between TAG and phospholipid molecules in the bilayer could occur spontaneously. In this process, it is more energetically favorable for TAG molecules to interact with each other than with phospholipid molecules, so TAGs may coalesce until the concentration exceeds a certain threshold level to form a “lens” [47]. Moreover, the composition of the ER membrane is not uniform, and thus demixing at some parts of the membrane can cost less energy. These conditions make it possible to ensure spontaneous demixing. Then some parts of the ER are highly bent to form protrusion tips. Although the mechanisms underlying this process is still unclear, protein–protein and/or protein–lipid interactions likely play important roles [5,48]. Several key proteins, including FIT proteins, SEIPINs, perilipins/PLINs, as well as acyl-CoA synthetases (ACSLs) and lipins, are proposed to regulate LD nucleation in mammals, while the proteins implicated in this process in plants are far less clear [49].

FIT proteins are exclusively localized to the ER. Mammals have two FIT proteins, FIT1 and FIT2. FIT2 has been confirmed to play an important role in LD formation [14,50]. Biochemical analyses confirmed that FIT2 directly and specifically binds to both TAG and its immediate precursor DAG, while purified FIT2 proteins have no intrinsic TAG biosynthetic activity. Because FIT proteins are enriched in the ER-LD region during lipid accumulation, it is believed that they help sequester TAG in the ER bilayer at the early stages of LD formation [2].

SEIPIN is an ER membrane protein which localizes at the ER–LD junction to determine the site of LD nucelation [8]. The protein contains two transmembrane segments, a highly conserved loop exposed to the ER lumen and two less conserved cytosolic termini [5]. In eukaryotic organisms, disruption of SEIPIN results in aberrant LD deposition in the cytoplasm [16]. In mammals, SEIPIN is enriched at ER-LD contact sites and is recruited to nascent LDs during LD biogenesis [51,52]. SEIPIN forms a homo-oligomeric, ring-shaped complex through interactions of its lumenal domains, consisting of eight β-strands and three or four short hydrophobic α-helices [53,54]. The LD phospholipid monolayers generally exhibit many packing defects which may be recognized by the conserved large hydrophobic side chains of the α-helices [55]. These helices and a helix at the N-terminal of the protein show a similar preference for LDs [53]. The eight β-strands in the SEIPIN lumen domain form a β-sandwich fold and may play a direct role in lipid sorting at the LD-ER border [16].

Unlike the single *SEIPIN1* gene found in humans and yeast, there are three *SEIPIN* homologs (*SEIPIN1-3*) in *Arabidopsis*, suggesting a more elaborate LD biogenetic machinery exists in plants [11]. Homology models show that the ER lumenal domain of each *Arabidopsis* SEIPIN displays a 3D structure similar to that of *Drosophila* and human SEIPIN, with few noticeable differences. Prediction of the quaternary structure shows that AtSEIPIN1 and AtSEIPIN2 assemble from 12 subunits, whereas AtSEIPIN3 forms an undecamer (11-mer). Protein sequences of the five SEIPINs from human, *Drosophila* and *Arabidopsis* are mainly different in the N-terminal regions, especially AtSEIPIN2 and AtSEIPIN3, which have much longer N-terminal regions than the others. The amphiphilic characteristics of the α-helices, the common motifs in different regions of ER lumenal domain were highly conserved [16]. Together, these suggests that plant SEIPINs may also have important homologous functions during LD nucleation, although how the three SEIPINs organize and whether a hetero-oligomeric complex is formed to control LD biogenesis is still unclear [56].

In *Arabidopsis*, the expression pattern of the SEIPINs coincides with the temporal activation of TAG biosynthesis during embryogenesis, indicating that SEIPINs play a role in LD biogenesis [57]. The sequence similarity of SEIPIN2 and SEIPIN3 is higher than that of SEIPIN1, and the N-terminal sequences of three proteins are substantially diverged [11,58]. *SEIPIN1* is expressed almost exclusively in developing embryos, while *SEIPIN2* and *SEIPIN3* were widely expressed across all tissues with maximal levels in embryos and pollen, where LDs accumulate most abundantly [57]. The expression of any of the three SEIPINs in yeast or plant (*Nicotiana benthamiana*) transient expression systems results in changing LD morphology. SEIPIN1 promoted the formation of larger-sized LDs, while SEIPIN2 and especially SEIPIN3 promoted small LDs [11]. The genetic analysis demonstrated that all three SEIPINs contribute to the biogenesis of LD in embryos, whereas only SEIPIN2 and SEIPIN3 play an important role in pollen, which is consistent with their coordinated expression in these tissues [59]. These results indicated that SEIPIN1 is functionally distinct compared to SEIPIN2 and SEIPIN3.

N-terminal deletions and swapping experiments of SEIPIN1 and SEIPIN3 revealed that the N-termini of SEIPINs play critical roles in the determination of LD size. In transiently expressing tobacco leaves, SEIPIN1 and SEIPIN3 are localized to the normal and discrete regions of the ER, specifically to ER-LD junctions where nascent LDs are being formed. Concomitantly, the normal, reticulated ER structure was reorganized into discrete domains that associated with LDs, indicating that SEIPINs participate in the organization of ER subdomains to support TAG synthesis and LD formation [11]. The results showed that plant SEIPINs have evolved specialized roles in the storage of neutral lipids by differentially modulating the number and sizes of LDs.

In the mature embryos of *seipin2*
*seipin3* double or triple *seipin* T-DNA mutants, LDs accumulated were significantly larger than those in wild-type embryos [59], similar to the phenotypes of *SEIPIN*-deficient mutants in animals and fungi [51]. The increased size leads to reduced packing efficiency of LDs in the *SEIPIN* mutants. Although the LD morphology changed in the *seipin* mutants, the TAG content in the mature seeds was not significantly affected [59,60]. These observations demonstrate that the role of SEIPINs is to stabilize ER-LD junctions to promote LD formation, rather than directly to promote the synthesis of TAGs.

Greer et al. found that the N-termini of SEIPIN2 and SEIPIN3 interact with the vesicle-associated membrane protein, (VAMP)-associated protein (VAP) family member AtVAP27-1 at ER-LD junctions [51,60]. AtVAP27-1 is involved in the formation of ER-plasma membrane contact sites (MCSs) in plants [61] and interacts with clathrin subunits to support endocytic trafficking [62]. The disruption of VAP27-1 results in the formation of aberrant, enlarged LDs in mature *Arabidopsis* seeds, similar to the phenotype observed in *seipin2seipin3* double mutants, suggesting AtVAP27-1 is important for the normal function of AtSEIPIN2/3 in plant cells.

The lipid droplet-associated protein (LDAP) is the LD ‘coat’ protein in nonseed tissues. Although first discovered in avocado mesocarp, LDAPs are broadly conserved in plants [63]. Three constitutively expressed *LDAP* genes (*LDAP1-3*) exist in *Arabidopsis*. Studies which ectopically expressed or disrupted the expression of any of the three *LDAP* genes have revealed that LDAPs are important for modulating LD number [64]. The LDAP-interacting protein (LDIP) was subsequently identified as a single-copy gene in *Arabidopsis*. *LDAP* and *LDIP* genes are both constitutively expressed in all organs examined during plant development, including in seeds. Disruption of LDIP leads to fewer, but larger LDs in leaves and seeds, with increases in the total amount of neutral lipids [64,65]. Expression and affinity-capture studies showed that LDIP associated not only with itself and LDAPs, but also with SEIPIN1-3 [65]. Recently, Pyc et al. found that LDIP works together with both SEIPIN and LDAP to facilitate LD formation in *A. thaliana* [66]. Co-expression of LDIP with SEIPIN (SEIPIN1 or SEIPIN2) resulted in a dramatic change in the subcellular distribution of LDIP, from LDs to the ER, thus LDIP was co-localized with SEIPIN at the ER. LDIP interacts with SEIPIN via a conserved hydrophobic helix in SEIPIN to modulate LD numbers and size. Thus, LDIP plays a key role in LD biogenesis via its interactions with both ER-localized SEIPINs, as well as LD localized LDAPs [66].

## 4. Growth of LD

The size of LDs varies greatly. Even within the same cell type, the size of LDs will change significantly with internal and external conditions [67]. LD growth can occur while it remains connected to the ER by a membrane neck and then buds from the ER at the final defined size, or nascent small LDs fuse to form large ones after budding. Which of these two possibilities is true may depend on the species and growth conditions [68,69].

In mammalian cells, the growth of LDs can be achieved either by increasing the amount of synthesized TAG or by the atypical fusion of LDs mediated by CIDE proteins [15]. Several enzymes, such as ACSLs, GPAT4, AGPAT3, Lipin1 and DGAT2, are reported to be localized to LDs and promote TAG synthesis and LD growth [9,70]. Two pre-droplets or nascent LDs might fuse when they laterally approach each other and are close enough. If the amount of TAGs in the different nascent LDs is the same, They will not influence each other. However, if the TAG amount in the nascent LDs is different, there will be a change in Laplace pressure. The higher internal pressure in the smaller LD will then drive the TAGs to flow from the smaller nascent LD to the larger one to form a fused LD [71]. CIDE proteins are enriched at LD contact sites (LDCS) and generate a potential pore (or channel structure) to allow lipid exchange between the contacted LD pair. Several proteins have been reported to promote LD fusion and unilocular LD formation. Perilipin (PLIN1) may stabilize the fusion complex by interacting with the fat-specific storage protein of 27 kDa (Fsp27, CIDE family protein) [72]. As an important regulator of LD fusion, Rab8a (a small GTP-binding protein) is involved in the formation of tight junctions between LD pairs [73].

Whereas plants have no direct CIDE protein FSP27 homologues, another possible mechanism of fusion between two nascent droplets during LD growth exists. The expression study in tobacco and yeast suggest that SEIPINs may also regulates LD growth in plants [11]. Another LD-associated protein, caleosin, may also have a role in LD nucleation and growth in plants. Caleosin was found to be involved in oil-accumulation [74]. Heterologous AtCLO1 expression in yeast causes accumulation of LD neutral lipids, resulting in larger and more abundant LDs containing more fatty acids and steryl esters [75]. The potential cis-elements in the caleosin gene promoter were analysed by using the web-based program PLACE, and the motif RAV1 was found out [58]. The transcription factor binding RAV1, containing AP2 and B3-like domain, is involved in synthesis of storage oil in castor bean [76].

## 5. LD Budding

Accumulation of TAGs leads to the formation of a flat lens between the two ER monolayers. Next, a nearly globular oil droplet surrounded by an ER monolayer is formed and then split from the mother membrane to release the LD. The membrane deformation process opreceeding LD release is called budding. The mechanisms underlying formation and detachment of budding LDs has been studied extensively in mammals, the nematode *Caenorhabditis elegans* and yeast. This process is mediated by several proteins, as well as surface interactions between the ER, the LD and the cytoplasm [16].

FIT proteins are essential for determining the directionality of the budding LD from the ER into the cytosol in mammals, nematode and yeast [2]. FITs directly and specifically bind to TAG, with the binding affinity correlating with LD size [77]. FIT2 is also found to be important for phospholipid metabolism and maintenance of general ER morphology. FIT2 displays lipid phosphate phosphatase (LPP) activity, which likely catalyzes PA to form DAG. Loss of this LPP activity leads to a dramatically altered ER morphology and defects in LD formation [78,79]. LD formation defects are likely secondary to membrane lipid abnormalities, possibly due to alterations in the phospholipids required for the LD coat [78]. Topology mapping has predicted that the active site of FIT2 is on the lumenal side of the ER membranes [14,79]. A model was hypothesized wherein the DAGs produced by FIT2 move or ‘flip’ to the outer leaflet membrane. Conversion of DAG back to a phospholipid would retain the glycerolipid species on the outer leaflet of the membrane. Therefore, FIT2 might be important for maintaining the phospholipid balance near LD biogenesis sites by promoting the transfer of phospholipids from the inner to the outer ER membrane, thereby increasing the availability of phospholipids for LD monolayer growth [80]. In yeast cells depleted of FIT2, the ER content of DAG increased and LDs remained embedded in the ER membranes and exposed to the ER lumen. Additional studies verified that non-bilayer phospholipids with negative molecular curvature, such as DAG or phosphatidylethanolamine (PE), favor the ER embedded state of LDs, while phospholipids with large positive molecular curvature, such as lysolipids, favor LD budding. Together, FIT2 plays a critical role in the budding of LDs, likely by modulating DAG levels at LD biogenesis sites [80].

During LD budding, SEIPIN found at ER-LD junctions appears to mark sites associated with the initial growth of nascent budding LDs [51,81]. The N-terminus of SEIPIN is required for directional LD budding into the cytoplasm [53,82]. Together with SEIPIN and FIT2, Perilipins/PLINs play a role in the directional LD budding from the ER membrane bilayer. PLINs are a family of abundant LD surface proteins which share the signature PAT domain and a conserved 11-mer amphipathic repeat segment that is important for targeting [83]. Five members (PLIN1-5) exist in mammals. Among which, PLIN3 binds to budding LDs early in the budding process by interacting with monolayer phospholipids [84]. Silencing of *PLIN3* attenuated LD maturation, suggesting PLIN3 plays a role in LD biogenesis [85]. Another possible candidate for LD budding is coat protein I (COPI). COPI is a heptameric complex and could bud off 60 nm nano-LDs from larger LDs in *Drosophila* cells in vitro [86,87,88].

In plants, however, the mechanism of LD budding has been poorly studied. No obvious homologs of FIT2 and PLINs are found in plants. Oleosin protein had been shown to play a role in LD budding [89]. Oleosins are the most abundant LD proteins in plant cells, and are important for the formation, stability and turnover of LDs. At least six oleosin lineages (P, U, SL, SH, T and M) have been identified and their evolutionary trajectory from green algae to advanced plants has been characterized [90]. Oleosin, like its homolog caveolin in mammals, has hairpin motifs and high positive curvature, which could naturally induce deformation of the ER outer monolayer for budding [91]. All oleosins have short amphipathic N- and C-terminal peptides located on both sides of the conserved 72-residue hydrophobic hairpin [92]. A previous study showed that accumulation of oleosins determines the LD size in *Arabidopsis* seeds [93]. OLE1 and OLE4 were found to negatively regulate LD size, whereas OLE2 might increase LD size [94]. Later expression of modified *Physcomitrella patens* oleosin genes in tobacco and *P. patens* have revealed that oleosins play a key role in the cytosolic directional release of ER-budding LDs. The whole hairpin of oleosin, including its entire length, the N-portion residues, as well as the three Pro and one Ser residues at the hairpin loop, are necessary for proper oleosin targeting and LD budding to the cytosol. Elimination of these necessary sites results in the modified oleosin entering the ER lumen and redirecting the LD to the ER lumen and then vacuoles [89]. Therefore, oleosin is the sole molecule identified in plants in vivo responsible for budding LDs into the cytosol.

Notably, electron microscopy analysis reveals that triple *seipin* mutant cells in *Arabidopsis* embryos can accumulate nuclear LDs (nLDs) attached to type I nucleoplasmic reticulum. This suggest that plants may also have an LD assembly domain at the nuclear membrane and SEIPINs are required to maintain the correct polarity of LD budding at the nuclear envelope, confining it to the outer membrane [59].

Although no obvious homologue of FIT2 has been found in plants, overexpressing mouse FIT2 in *Arabidopsis* dramatically increased LD number, size and TAG accumulation in both leaves and seeds. In transiently expressed tobacco leaves, FIT2 localized exclusively to the ER and was often concentrated at ER-LD junction sites. When co-expressed with *Vernicia fordii* DGAT2 and *Arabidopsis* SEIPIN1, FIT2 colocalized with both proteins, consistent with the premise that FIT2 associates with ER domains involved in both TAG biosynthesis and nascent LD biogenesis in plants. Mouse FIT2 likely interacts with the native LD biosynthetic machinery in plant cells to promote enhanced TAG compartmentalization and LD budding. In addition, the role of FIT2 in LD biogenesis is likely dependent on the biophysical properties of cellular membranes that might also be present in plant cells. These results indicate that there is a surprising functional conservation in higher plant cells which do not contain FIT2 homologues. Other proteins in plants might provide an analogous function [39].

Indeed, LPPs and lipins that bind to the ER do exist in plants, and the deficiency of these proteins leads to serious changes in ER membrane organization [27,95,96]. Plants also have phospholipid flippases, which may be involved in the exchange of phospholipids between the inner and outer leaflets of the ER membranes [97]. However, whether these proteins play a role in LD biogenesis, beyond the role of phosphatase in providing DAG for TAG biosynthesis, remains to be investigated [16].

The corresponding homologs of the COPI subunits have also been identified in plants cells [98]. In plants, COPI is involved in retrograde transport from the Golgi to the ER, Golgi maintenance and cell-plate formation, and its prolonged depletion induces programmed cell death in leaf cells [99]. A recent study demonstrated that COPI may contribute to the acceptance of compatible pollen grains [100]. In liverwortare, analysis of a *Marchantia polymorpha* T-DNA-mutant with LDs with increased circularity confirmed that COPI subunit MpSEC28 is responsible for this phenotype and that normal LD formation requires COPI-mediated secretory activity [12]. However, it is unclear whether COPI plays a role in LD budding in other plant species.

It is worth noting that the topology of LDs is also unusual. All other cellular organelles are encapsulated by a bilayer, while LDs are surrounded by a phospholipid monolayer. The molecular interactions between oil molecules and phospholipids of the ER membrane are also able to mediate LD budding without any energy input from proteins [101]. The position of LDs between the ER and cytoplasm can be envisaged as three different phases with different surface tension, namely the cytosol, the ER lumen and the oil phase. In vitro studies of artificial LDs from model membranes suggest that the phospholipid composition of the ER membrane and the membrane surface tension are key parameters controlling LD budding and releasing into the cytoplasm in cells [102,103].

## 6. LD Coat Formation

LDs are coated by a monolayer of surface polar lipids, thus creating a large amphipathic interface for specific proteins to insert into [104]. In seed plants, the major proteins on LDs are oleosin, caleosin and steroleosin [105]. The plant-specific LDAPs and LDIP are LD ‘coat’ proteins and may govern the dynamics of LDs in specialized non-seed tissues [63,65]. A few surface proteins of algal LDs, such as major LD protein (MLDP) [106], Haematococcus oil globule protein (HOGP) [107], lipid droplet surface protein (LDSP) [108], caleosin-related Symbiodinium LD protein (SLDP) [109] and Stramenopile-type lipid droplet protein (StLDP) [110], have also been identified.

Oleosin proteins are the major LD coat proteins in seeds. During soybean seed development, about 95% of oleosin was targeted to LDs, with the remaining 5% being located on the ER near LDs, suggesting oleosins target LDs via the ER in vivo [111]. Oleosins can stabilize artificial LDs, suggesting that they are capable of spontaneous insertion into the TAG matrix [112]. Suppression of oleosin produces the formation of fewer, larger LDs in seeds, likely due to LD-LD fusion [94]. Oleosins are cotranslationally synthesized on the cytosolic side of the ER via signal-recognition particle (SRP)-guided mRNAs [113,114]. Newly synthesized oleosins are migrated towards sites of TAG accumulation and inserted into the ER [115]. The central hydrophobic region of the protein is required for targeting and/or membrane insertion, while the N- and the C-terminal hydrophilic domains have been found to be important to dictate the correct topology of the protein within the ER membrane in vitro. Moreover, there are several domains in oleosin which interact with the SRP to direct the protein to the ER membrane [114]. In animal cells, the binding of PLIN to the LD surface promotes the formation of the LD coat and LD growth [116]. No obvious homologs of the PLIN proteins have been found in plants, so oleosins may play an analogous role in forming the LD coat in developing seeds.

LDAPs and LDIPs are ubiquitously expressed in *Arabidopsis* and influence LD abundance in leaves [64,117]. The hydrophilic N-terminal of LDIP, containing a discrete amphipathic α-helical targeting sequence, is necessary and sufficient for targeting the LD surface [65]. Loss of LDIP results in a significant change of LD number and size in leaves and seeds when the predominant LD coat proteins (i.e., LDAPs and oleosins) are still present. A recent study confirmed that LDAP binds first to LDs and then recruits LDIP to the LD surface. These observations indicated that LDIP is important for regulating LD number and size in a manner independent of, and perhaps functions upstream of, the LDAPs and oleosins [66].

Oleosins are present in seeds, pollen grains, and the floral tapetum of some plant species, but are absent in photosynthetic tissues [118]. The photosynthetic, vegetative cells, such as the unicellular green alga *Chlamydomonas reinhardtii*, can also form cytosolic LDs. MLDP, as an abundant protein on LD, is mainly associated with the LD surface while a subfraction is associated with the ER-LD connection region. Unlike oleosin, MLDP has no long hydrophobic segment but is more hydrophobic than oleosin [106,119]. Reducing the amount of MLDP by RNA interference (RNAi) results in the formation of larger LDs. Depending on the availability of nitrogen, MLDP recruits different proteins to the LD surface, especially tubulin. On the contrary, the depolymerization of microtubules by using the drug colchicine disturbed the association between MLDP and LDs, suggesting functional microtubules may play a role in directing MLDP to LDs. Taken together, MLDP acts as a center for protein recruitment to stabilize mature LDs [120].

The abundance of LDSP in heterokont algae *Nannochloropsis* cells closely parallels the TAG content change. Expression of LDSP in an *Arabidopsis oleo1* mutant showed reduced LD size and prevented LD coalescence. These results indicate that LDSP plays an important role in the formation and stabilization of LDs in ways partially analogous to plant oleosin [108]. As a main surface protein from *Phaeodactylum tricornutum*, StLDP expression levels were also found to be correlated with a change in LD size [110].

Among the proteins associated with LDs, only oleosin, LDSP and StLDP have central hydrophobic domains (72 and 40–60 residues, respectively) long enough to penetrate the TAG core and stabilize the LD. Others have short or no hydrophobic fragments and may interact loosely or transiently with the LD surface.

The LD monolayer includes various proteins with different functions depending on the needs of the cell and the organism. However, how these specific proteins move from the ER to the LD surface is still unclear. The Arf1/COPI protein is a protein complex involved in the movement of vesicles around cells and may recruit proteins to the LDs. Wilfling et al. proposed that Arf1/COPI act directly on LDs in *Drosophila* cells to decrease the concentration of surface phospholipids through the budding of nano-LDs [88]. Subsequently, LD surface tension increases and the formation of LD-ER membrane bridges are triggered for rapid membrane-associated targeting to LDs, such as ATGL (major TAG lipase) [121] and GPAT4. However, the role of plant COP1 in the process was still not found.

In an experiment with *Arabidopsis* pollen tubes, Sec61g, a subunit of the ER protein translocation machinery (Sec61 translocon complex), was isolated. In transient expression systems, this protein localizes to regions of the ER that in some instances were in close proximity to or encircled LDs. The Sec61 complex is important for the incorporation of nascent proteins into the ER [122] and might also function at ER-LD junctions [11]. Ectopic expression of sunflower oleosins in yeast demonstrated that Sec61p, the homolog of Sec61 in yeast, is required for ER membrane-insertion of oleosin [114]. Therefore, this complex may also participate in the targeting of LD proteins [123].

## 7. Other Proteins and Factors Involved in LD Formation

### 7.1. Rab Proteins and Sterols Involved in LD Formation

Recent evidence suggests that Rab proteins are involved in LD formation [124]. Rab proteins are small membrane-related GTP-binding proteins that are involved in almost all stages of vesicular transport, including vesicle budding and delivery, tethering of the vesicle membrane and target compartment fusion. These proteins cycle between GDP-binding (inactive) and GTP-binding (active) states, which relies on guanine nucleotide exchange factor proteins (GEFs), and transmits upstream signals to downstream effectors [125]. More than 30 Rab proteins have been found to associate with LDs in mammalian systems [126]. Several Rab proteins, such as Rab18, Rab1, Rab40c and Rab8a, may play roles in LD biogenesis [73,127,128,129]. Rab18 is the best characterized Rab family protein and directly associates with the LD surface. Rab18 deficiency results in a dramatic decrease in LD number and increase in LD size [130]. In *Drosophila* cells, Rab18 recruits the ER-associated soluble N-ethylmaleimide-sensitive factor activating protein receptors (SNARE) and NAG-RINT1-ZW10 (NRZ) complex near to the LDs and tether the ER to LDs through Rab18-NRZ-SNARE interactions [131,132]. This process facilitates lipid incorporation from the ER to LDs and promotes LD growth.

However, the direct involvement of Rab proteins in plant LD biology has yet to be investigated. In tobacco seedlings and pollen tubes, homologs of Rab18 was enriched in LD fractions [133]. In young stomata of *Arabidopsis*, RABC1, the homolog of mammalian Rab18 localizes to the ER and weakly localizes to LDs under normal conditions. However, in an RABC1 GEF1-dependent manner, RABC1 mainly targets the LD surface in response to oleic acid application, which is known to stimulate LD formation. Further studies showed that RABC1 physically interacts with SEIPIN2/3 and the interaction depends on the RABC1 activity state. Deficiency of RABC1, RABC1GEF1, or SEIPIN2/3 resulted in aberrantly large LDs and aberrant LD dynamics in stomatal lineage cells. When LD production is induced by oleic acid, SEIPIN2/3 is mainly located on the LD surface in wild-type, while very little SEIPIN2/3 was located on the LD surface in *RABC1* mutants. Thus, together with its partner GEF, RABC1 negatively controls the dynamics and size of LDs in young stomata by regulating the localization of its effector factors SEIPIN2/3 [13].

Recently, Yu et al. [134] found that sterols also play a role in the formation of oleosin-coated LDs in *Arabidopsis*. The *dwf5-10/OLE1* mutant, lacking a sterol Δ7-sterol-C5-desaturase, was almost completely devoid of leaf LDs and TAG. The mutant also displayed reduced LD number, increased LD size and reduced oil content in seeds. Upon oleic acid feeding, leaves of the mutants accumulated large-sized individual LDs lacking oleosin. Deficiency in 24-ethyl-Δ^5^-sterols and 24-ethylidene-Δ^5^-sterol is responsible for the phenotype of the mutants. The data revealed that membrane lipids play an important role in LD biogenesis by coordinating the synthesis of oil and oleosins and their assembly into LDs.

### 7.2. Autophagy Plays a Role in LD Formation

Previous studies in animals has revealed that autophagy is implicated in the accumulation of LDs [135]. The role of autophagy in the formation of plant LDs is unclear. Some indirect evidence comes from a study on the role of autophagy in male reproductive development in rice. The LD amounts in tapetal cells, and the levels of TAGs and DAGs in pollen grains were significantly lower in the autophagy-deficient mutant *Osatg7* than in the wild type, suggesting autophagy might be required for the formation of tapetal LDs [136].

The unicellular algae *C. reinhardtii* cells accumulate high levels of TAGs and large numbers of LDs when subjected to nitrogen deprivation. The increased accumulation of TAG and LDs was fully suppressed by the vacuolar ATPase inhibitor concanamycin A, a reagent known to inhibit autophagy. These results suggest that TAG biosynthesis and LD formation require autophagy under nutritional stress [137].

All cell types maintain two different levels of autophagy. The basal level of autophagy maintains cellular homeostasis under favorable growth conditions, whereas the massive inducible level of autophagy functions primarily as an adaptive response to many developmental and environmental stresses [138,139,140]. Fan et al. [141] demonstrated that basal autophagy contributes to TAG synthesis and LD biogenesis in mature and senescing leaves of adult *Arabidopsis* plants, whereas inducible autophagy contributes to LD degradation. The study revealed a dual role for autophagy in regulating lipid synthesis and turnover in plants.

## 8. Conclusions and Future Perspectives

LDs are ubiquitous in most organisms, including in animals, protists, plants and microorganisms. They are dynamic organelles that play key roles in various aspects of plant physiology, in addition to energy storage. During the past decade, the research of LD formation, a very complex and precisely regulated process, has been a focus of intensive research. The research has mainly focused on the identity of the specific proteins involved in LD biogenesis. Although emerging experimental evidence revealed that LD biogenesis is a spatiotemporally regulated process, occurring at specific sites of the ER defined by a specific set of lipids and proteins, the exact mechanism of this process is still unclear. Moreover, the evidence obtained is mostly from animals and yeast, whereas to date the information about LD biogenesis and proteins involved in the process in plants is more scarce (Table 1). Moreover, the ways in which LDs form in plant cells likely includes several mechanisms that are cell type- or stage-specific [88,105].

In the future, many interesting questions in plants, such as those relating to the identification of homologous proteins of the newly characterized LD formation regulatory proteins and proteins targeting to the surfaces of LDs, the behavior of lipids or proteins during LD formation, and the mechanistic details of LD biogenesis, will require further investigation. The results of these studies could lay the foundation for further establishing a complete and accurate model of the mechanisms underlying LD biogenesis.

## Figures and Tables

**Figure 1 ijms-24-07476-f001:**
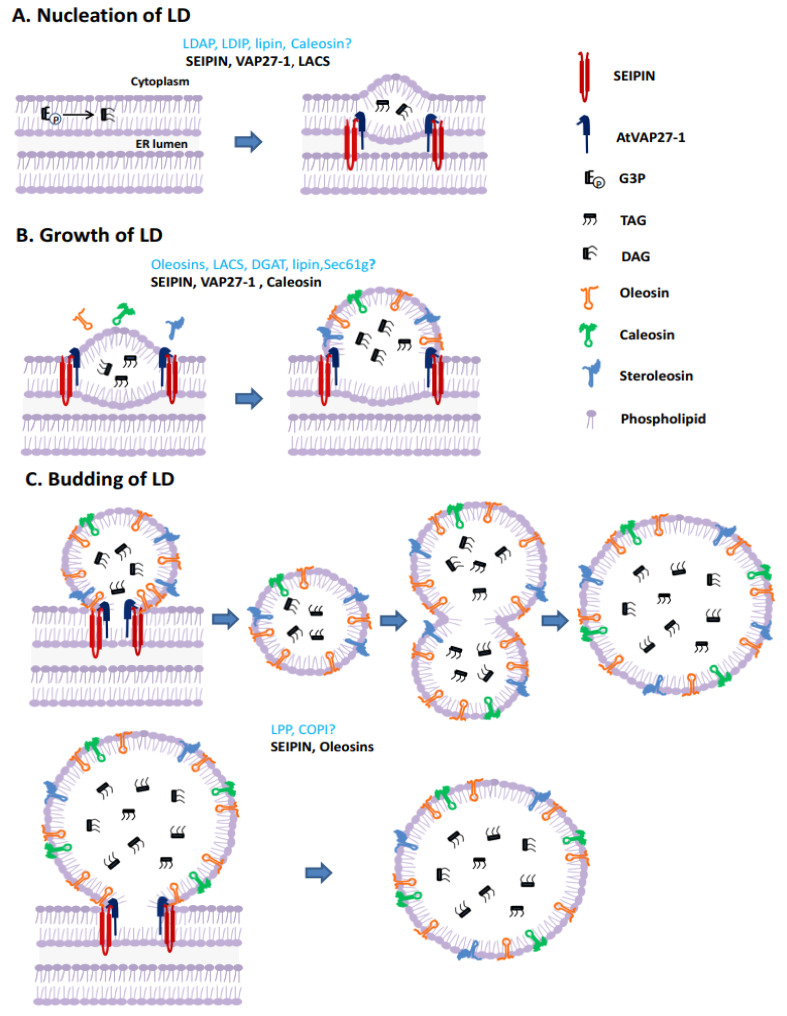
Models of lipid droplet biogenesis in plant (using LDs in seeds as an example). (**A**) Nucleation of LD. Neutral lipids (mainly TAG) nucleate and a lens-like structure is formed in the ER membrane. (**B**) Growth of LD. LD-specific proteins synthesized in the cytoplasm migrate toward sites of TAG accumulation and are inserted into the ER. The size of the neutral lipid core increases and protrudes due to the accumulation of TAG. (**C**) LD budding. After reaching a critical size, a nearly globular LD surrounded by an ER monolayer is formed and then splits from the mother membrane. Alternatively, nascent small LDs fuse to form large ones after budding.

**Table 1 ijms-24-07476-t001:** Identified genes and proteins involved in LD biogenesis in *Arabidopsis thaliana*.

Gene	Protein Name	TAIR Locus	Localizati on	Putative Function	Reference
*FATA*	Fatty acyl-ACP thioesterase A	At3g25110.1	Plastid	FA biosynthesis and export from plastid	[18]
*FATB*	Fatty acyl-ACP thioesterase B	At1g08510.1	Plastid	FA biosynthesis and export from plastid	[18]
*FAX2*	Fatty acid export 2	At2g38550	Cytoplasm	FA biosynthesis and export from plastid	[10]
*FAX4*	Fatty acid export 4	At1g33265	Cytoplasm	FA biosynthesis and export from plastid	[10]
*LACS1*	Long-chain acyl-CoA synthetase 1	At2g47240	Cytoplasm	FA biosynthesis and export from plastid	[10]
*LACS9*	Long-chain acyl-CoA synthetase 9	At1g77590.1	Cytoplasm	FA biosynthesis and export from plastid	[10]
*GPAT9*	Glycerol-3-phosphate acyltransferase 9	At5g60620.1	ER	Acyl-CoA dependent TAG synthesis	[22]
*LPAAT2*	Lysophosphatidic acid acyltransferase	At3g57650.1	ER	Acyl-CoA dependent TAG synthesis	[24]
*AtPAH1*	Phosphatidic acid phosphohydrolase1	At3g09560	ER	Acyl-CoA dependent TAG synthesis	[25]
*AtPAH2*	Phosphatidic acid phosphohydrolase2	At5g42870	ER	Acyl-CoA dependent TAG synthesis	[25]
*DGAT1*	Diacylglycerol O-acyltransferase 1	At2g19450.1	ER, LD	TAG synthesis in developing seeds	[28]
*DGAT2*	Diacylglycerol O-acyltransferase 2	At3g51520.1	ER, LD,	Unusual fatty acid formation	[28]
*DGAT3*	Diacylglycerol O-acyltransferase 3	At1g48300.1	Cytoplasm	Regulation of the acyl-CoA pool size and composition	[28]
*PDAT1*	Phospholipid-diacylglycerol acyltransferase	At5g13640.1	ER	Acyl-CoA independent pathway	[38]
*SEIPIN1* *SEIPIN2* *SEIPIN3*	SEIPIN 1 SEIPIN2 SEIPIN 3	At5g16460, At1g29760, At2g34380	ER-LD junctions	Nucleation and growth of LD, LD budding	[11]
*LDAP1* *LDAP2* *LDAP3*	LD-associated proteins	At1g67360 At2g47780 At3g05500	LD	Nucleation and growth of LD in leaves, LD coat protein	[64]
*LDIP*	LDAP-interacting protein	At5g16550	LD	Nucleation and growth of LD, LD coat protein	[65]
*VAP27-1*	Vesicle-associated membrane protein-associated protein	At3g60600	ER-PM junction	Nucleation and growth of LD	[61]
*CLO1*	Caleosin1	At4g26740	LD	Nucleation and growth of LD	[75]
*OLE1*	Oleosin1	At4g25140	ER, LD	Negative regulator of LD size, LD budding, LD coat protein	[92]
*OLE2*	Oleosin2	At5g40420	LD	Negative regulator of LD size	[94]
*OLE4*	Oleosin4	At3g01570	LD	Postive regulator of LD size	[94]
*COPI*	Coat protein I complex		Golgi	Unknown	[99]
*EC61g*	SEC translocase subunit	At5g50460	ER-LD junction	May participate in the targeting of LD proteins	[123]
*RABC1*	RABC1	At1g43890	ER, LD	Negative regulator of LD size	[13]
*DWF5*	Δ7-sterol-C5-desaturase	At1g50430	Cellular membrane	LD biogenesis	[134]

## Data Availability

Not applicable because the paper is the opinion based on the analysis of the piublished literature.

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
