# Peer review of "Dynamic Regulation of Lipid Droplet Biogenesis in Plant Cells and Proteins Involved in the Process"

_ijms, 2023, doi:10.3390/ijms24087476_

Round 1

Reviewer 1 Report

Article titled: Dynamic Regulation of Lipid Droplet Biogenesis in Plant Cells and Proteins Involved in the Process

The current review has a good objective, and as it is known characterized the Dynamic Regulation of Lipid Droplet Biogenesis in Plant Cells and Proteins Involved in the Process. work on the molecular processes that control LD formation in plant cells and highlight the proteins that govern this process, hoping to provide useful clues for future research. to complete the objective of the current study

- Abstract:
-it is good, but it short, the authors should consider the proposed changes for improving the clarity of the content as  The mechanisms underlying LD biogenesis had remained incompletely

Keyword: good

-Introduction part is appropriate but a few things are needed for further improvements especially the study aims should be added. Update the references

Need details about mod of action of the physiological regulation mechanisms of Regulation of Lipid Droplet Biogenesis in Plant Cells gene is overexpressed

Add some studies about the study with highlighting research gaps, which necessitated conducting this trial.

Conclusion: good 

References:
-Cross-check the references in the text and reference cite. Few references are not as per journal style in the text as well reference section

-

Reviewer 2 Report

Dear Corresponding Author
I checked your paper and I can suggest below sentences to you:
Writing a review paper is not gathering some sentences and compile them. For doing a review you will need to have a strategy to mange the story of the subject that you want to review it.
In the main text, I could not find recent works (2023, 2022, 2021, ...) related to the title of the paper. There are some English errors and you will need to request a native Englishman to edit it.
Nothing is in the format of the MDPI journal.
Therefore, I think authors were in a hurry to write and submit the paper that cause decreasing of quality of their paper.
I encourage the to re-evaluate the paper carefully and update it and then resubmit it as a new submission.
With Best Regards

Round 2

Reviewer 2 Report

For the moment, your paper is better. After correcting of some format and English issues it is OK for publication.